# Boosting-Based Classifiers for Retinal OCT Disease Detection: A Multifractal Feature Approach

**Ahlem Aziz** [1]**, Necmi Serkan Tezel** [1]**, Youcef Attallah** [2]

[1] Electrical and Electronics Engineering Department, Karabuk University, 78050 Karabuk, Türkiye
[2] Department of Electronics, University of Science and Technology of
Oran Mohamed-Boudiaf (USTOMB), 31000, Oran, Algeria
ahlem98aziz@gmail.com

## Abstract

Diabetic retinopathy (DR) and other retinal diseases remain leading causes of preventable vision loss worldwide, stressing the need for reliable and automated screening systems. Optical Coherence Tomography (OCT) provides detailed cross-sectional views of the retina, enabling quantitative analysis of structural changes. In this study, we employ multifractal feature extraction from OCT images to characterize retinal tissue irregularities. This study focuses on a comparative evaluation of boosting-based classifiers, namely Gradient Boosting, XGBoost, and LightGBM. Experimental results demonstrate that Gradient Boosting achieved an accuracy of $94.97\%$, XGBoost reached $95.27\%$, and LightGBM obtained the highest accuracy of $95.58\%$. These findings highlight the effectiveness of boosting algorithms in retinal OCT image classification, confirming LightGBM as a promising approach for accurate and efficient automated screening tools in ophthalmology.

## 1   Introduction

Diabetic retinopathy (DR), a microvascular complication of diabetes, is one of the leading causes of preventable blindness in working-age adults worldwide [1]. Because early stages of DR are often asymptomatic, timely diagnosis is a major clinical challenge, as undetected disease can lead to irreversible vision loss [2]. Imaging modalities such as fundus photography, fluorescein angiography, Optical Coherence Tomography (OCT), and OCT Angiography (OCTA) are widely employed in clinical practice [3]. Among them, OCT has emerged as a particularly powerful and non-invasive tool, providing high-resolution cross-sectional images of retinal microstructures and enabling the detection of subtle pathological changes before they become clinically visible [4], [5].

The retinal vasculature exhibits irregular, self-similar branching patterns that cannot be fully captured by classical Euclidean geometry. Fractal and multifractal analysis provide mathematically rigorous frameworks to quantify such structural complexity [6], [7]. Unlike a single fractal dimension, multifractal descriptors characterize local heterogeneity in vascular and tissue structures, capturing subtle alterations associated with early DR progression [8]–[11]. When applied to OCT images, multifractal analysis highlights textural and morphological changes that may not be detectable with conventional image processing techniques. In parallel, recent advances in machine learning (ML) have transformed medical image analysis, offering automated and scalable solutions for retinal disease detection [12]–[14]. Several studies have demonstrated the benefit of combining multifractal descriptors with supervised learning algorithms for DR classification [15]–[18].

Although multifractal descriptors provide valuable biomarkers for retinal tissue characterization, their practical deployment in automated DR detection systems faces significant challenges. Traditional classifiers often fail to capture the complex, non-linear decision boundaries inherent in such high-

Submitted to 39th Conference on Neural Information Processing Systems (NeurIPS 2025). Do not distribute.

dimensional features. Furthermore, balancing high diagnostic accuracy with computational efficiency remains an unresolved issue, particularly for clinical applications where real-time analysis is essential.

Despite promising results, prior works have often relied on limited classifier architectures or lacked systematic evaluation of modern ensemble methods. In particular, the comparative effectiveness of boosting-based classifiers on multifractal descriptors derived from OCT images remains underexplored. Moreover, the need for lightweight yet accurate models suitable for integration into clinical workflows has not been sufficiently addressed.

This paper addresses these gaps by conducting a comparative study of boosting-based classifiers—namely Gradient Boosting, XGBoost, and LightGBM—trained on multifractal descriptors extracted from the Retinal OCT Image Classification – C8 (2021) dataset. The objective is to evaluate their relative performance, assess generalization ability, and identify efficient classifiers that can be deployed in automated OCT-based screening systems.

The remainder of this paper is structured as follows. Section 2 details the proposed methodology, including the multifractal feature extraction process and the implementation of boosting-based classifiers. Section 3 presents and critically discusses the experimental results, with emphasis on comparative performance analysis. Finally, Section 4 concludes the study and outlines future research directions aimed at enhancing automated OCT-based retinal disease screening.

## 2    Methodology

The proposed methodology integrates multifractal feature analysis with boosting-based classification models for automated retinal disease detection using Optical Coherence Tomography (OCT) images. OCT provides high-resolution cross-sectional visualization of retinal layers, offering a non-invasive imaging modality that is widely available in clinical practice and well suited for early pathological screening [19].
To extract discriminative information, multifractal analysis is employed, as it effectively characterizes the irregular geometric and structural patterns present in retinal tissue. The extracted descriptors capture variations in local and global complexity, providing a robust representation of retinal morphology that is not readily observable through standard pixel- or intensity-based analysis [9], [20].
The flowchart outlines the proposed pipeline. OCT images are first preprocessed, including resizing to 100×100 pixels and segmentation using CLAHE, Gaussian blur, and thresholding. The enhanced images are then subjected to multifractal analysis, followed by feature extraction. These features are classified with a Gradient Boosting model, and performance is evaluated using accuracy, precision, recall, and F1-score (*Figure 1*).

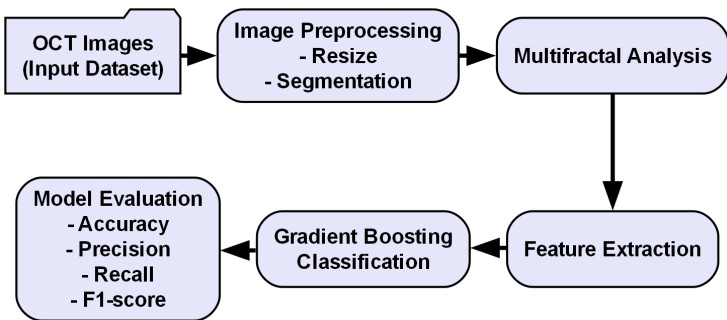

Figure 1: Flowchart of the proposed methodology

Following feature extraction, three ensemble-based classifiers—Gradient Boosting (GB), Extreme Gradient Boosting (XGBoost), and Light Gradient Boosting Machine (LightGBM)—are applied. These algorithms are particularly well suited for structured biomedical data, as they combine high predictive capacity with computational efficiency and scalability. A K-fold cross-validation framework is adopted to ensure rigorous performance evaluation and to reduce the risk of overfitting [21].

## 2.1 OCT Dataset

This study employs the publicly available *Retinal OCT Image Classification – C8* dataset, which contains approximately 24,000 high-resolution OCT images categorized into eight distinct retinal conditions. The images were sourced from multiple open-access repositories, including Kaggle and OpenICPSR, and subsequently preprocessed to ensure class balance and consistent distribution [22]. For the present work, two categories—*Normal* and *Diabetic Retinopathy*—were selected to define a binary classification task targeting early DR detection. Each class contained approximately 3000 images, ensuring balanced representation across categories. The heterogeneous origin of the dataset introduces variability in both imaging conditions and pathological presentations, thereby supporting the development of models with improved generalizability. Nevertheless, the dataset lacks detailed clinical metadata (e.g., patient demographics, disease stage), which constrains opportunities for stratified analysis. While sufficient for the experimental scope of this study, future validation on clinically annotated cohorts will be essential to establish translational reliability and clinical applicability [22].

## 2.2 OCT Image Preprocessing

The first stage of the proposed pipeline consists of OCT image preprocessing, implemented using the open-source Fiji platform (ImageJ distribution) with a custom macro script [19]. This step is essential for improving image quality and preparing the data for subsequent feature extraction. Contrast Limited Adaptive Histogram Equalization (CLAHE) was applied to enhance local contrast and reveal subtle retinal structures, while Gaussian blur was employed to suppress high-frequency noise and smooth image textures. Following this, thresholding-based segmentation was performed to delineate and isolate relevant retinal regions, ensuring that diagnostically meaningful structures were preserved for subsequent multifractal analysis [23].

## 2.3 Multifractal Analysis

Multifractal analysis provides a mathematical framework for characterizing structures that exhibit non-uniform scaling behavior across multiple spatial resolutions. In medical imaging, and particularly in OCT scans, this approach enables the quantification of tissue heterogeneity and structural irregularities, which may serve as indicators of pathological changes associated with diabetic retinopathy [24]. For this study, multifractal characterization was performed using the *Multifrac v1.0.0* plugin, an ImageJ extension designed for multifractal assessment of both two- and three-dimensional image data. The software is publicly available via the official ImageJ distribution platform [25]. The plugin computes multifractal descriptors via the box-counting algorithm, a widely used method for estimating fractal dimensions. It also provides integrated preprocessing options such as resizing, binarization, and scale selection, while supporting automated analysis with structured output of both numerical results and processed images. Full technical details are available in the documentation and original publication by Torre et al. (2020) [26].

In the present work, two-dimensional multifractal analysis was applied to the preprocessed OCT images. After binarization, the analysis was restricted to white pixels to highlight retinal structures of clinical relevance. The resulting feature set captures the spatial complexity and scaling behavior of retinal textures. The principal descriptors extracted were:

- **Generalized dimensions:**

    - *Box-counting dimension* ($D_B$): quantifies overall scaling density.
    - *Information dimension* ($D_I$): measures the distribution of intensity probabilities.
    - *Correlation dimension* ($D_C$): emphasizes local spatial correlations.

- **Singularity spectrum** ($f(\alpha)$): characterizes the distribution of local scaling exponents, thereby capturing the multifractal diversity of the retinal image.

These multifractal descriptors form the feature vector used in the subsequent classification stage. The following subsection elaborates on the mathematical definitions and interpretative value of each parameter.

## 2.4 Multifractal Feature Extraction

Multifractal analysis provides a principled framework for describing scale-invariant properties in OCT images by quantifying structural irregularity and heterogeneity. In this study, two categories of descriptors were extracted: generalized dimensions and the singularity spectrum.

### 2.4.1 Generalized Dimensions

The generalized dimensions $D_q$ describe how probability measures scale across different orders $q$ [27]:

$$D_q = \frac{1}{q-1} \lim_{\epsilon \to 0} \frac{\log \sum_i p_i^q(\epsilon)}{\log \epsilon}, \tag{1}$$

where $\epsilon$ is the box size and $p_i(\epsilon)$ the normalized measure in the $i$-th box.

For specific values of $q$, $D_q$ yields clinically meaningful descriptors:

- **Box-counting dimension** ($D_B$, $q = 0$):

$$D_B = \lim_{\epsilon \to 0} \frac{\log N(\epsilon)}{\log(1/\epsilon)} \tag{2}$$

  with $N(\epsilon)$ the number of non-empty boxes. $D_B$ reflects global geometric complexity and can highlight microstructural degradation in DR.

- **Information dimension** ($D_I$, $q = 1$):

$$D_I = \lim_{\epsilon \to 0} \frac{\sum_i p_i(\epsilon) \log p_i(\epsilon)}{\log \epsilon} \tag{3}$$

  capturing heterogeneity in intensity distributions, relevant to irregular deposits and localized edema.

- **Correlation dimension** ($D_C$, $q = 2$):

$$D_C = \lim_{\epsilon \to 0} \frac{\log \sum_i p_i^2(\epsilon)}{\log \epsilon} \tag{4}$$

  emphasizing local clustering of intensities, often linked to lesion aggregation and structural disruptions in DR.

### 2.4.2 Singularity Spectrum

The singularity spectrum $f(\alpha)$ characterizes the distribution of local singularities and provides a compact description of textural variability [28]:

$$\alpha(q) = \frac{d}{dq}[(q-1)D_q], \tag{5}$$

$$f(\alpha) = q\alpha - (q-1)D_q, \tag{6}$$

where $\alpha$ is the Hölder exponent and $f(\alpha)$ the fractal dimension of the set of points with singularity strength $\alpha$.

Key descriptors include:

- $\alpha_{\min}$: strongest irregularities (e.g., hemorrhages, exudates).
- $\alpha_{\max}$: smoothest regions (healthy retinal layers).
- $\alpha_{\text{center}}$: dominant singularity type.
- $f(\alpha)_{\max}$: abundance of dominant structural patterns.
- Spectrum width $\Delta\alpha = \alpha_{\max} - \alpha_{\min}$: degree of multifractality, increasing with pathological variability.
- Symmetric shift: $\alpha_{\text{center}} - \frac{\alpha_{\min} + \alpha_{\max}}{2}$, indicating texture asymmetry toward finer or coarser structures.

Clinically, $\alpha_{\min}$ often corresponds to dense pathological regions (microaneurysms, neovascular tufts), while $\alpha_{\max}$ highlights smooth tissue zones. Progression from early to advanced DR is reflected in a broader spectrum ($\Delta\alpha$) and increased asymmetry, denoting heterogeneous pathological remodeling.

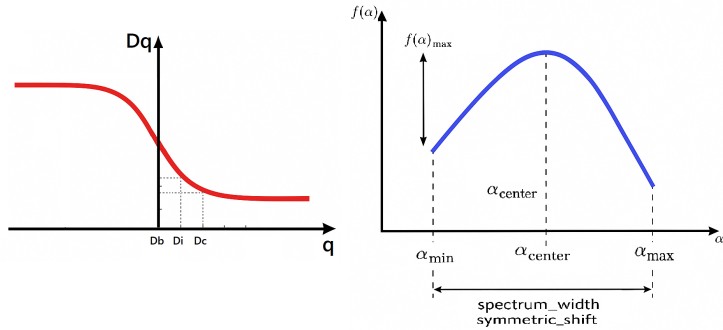

Figure 2: Generalized Dimension Curve $D_q$ and Multifractal Spectrum $f(\alpha)$

## 2.5 Gradient boosting variants

Gradient boosting is an ensemble learning technique used for classification and regression. It combines multiple models to improve overall performance. The main idea is to build a sequence of weak learners—models that perform slightly better than random guessing, usually decision trees. Each new tree corrects errors made by the previous ones. While bagging trains models in parallel, boosting builds the model in stages. This iterative, weighted construction progressively improves performance [29].

Gradient Boosting (GBM) serves as the basic method in this family of models. In each iteration, it minimizes a differentiable loss function—a mathematical measure of error that is optimizable using calculus—using gradient descent, which iteratively updates parameters to find the minimum value of the loss. This approach enables each new tree to target the residual errors (the differences between actual values and predicted outcomes) from prior trees in the ensemble. While GBM is effective, it can experience lengthy training times and has limited scalability with large datasets [29].

To address these issues, Extreme Gradient Boosting (XGBoost) was introduced as an optimized implementation of GBM. It incorporates several enhancements, including L1/L2 regularization to mitigate overfitting, efficient memory management, and parallelized algorithms to expedite training. These improvements have made XGBoost one of the most widely adopted models in machine learning competitions [30].

Similarly, Light Gradient Boosting Machine (LightGBM), developed by Microsoft, further advances the efficiency of gradient boosting. It is designed specifically for speed and scalability. LightGBM introduces two main innovations: Gradient-based One-Side Sampling (GOSS), which selects only the most significant data points during training to reduce computational cost; and Exclusive Feature Bundling (EFB), which groups features that do not take nonzero values simultaneously, reducing the number of features considered. These techniques help LightGBM efficiently handle large and high-dimensional datasets while maintaining accuracy [29].

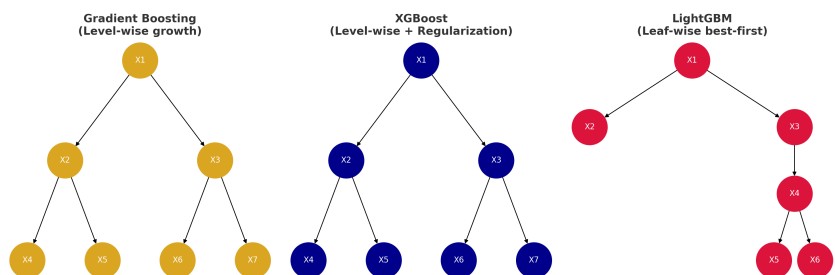

Figure 3: Structural Differences Between Gradient Boosting, XGBoost, and LightGBM

## 3 Results and Discussion

### 3.1 Experimental setup and hyperparameters

To ensure a fair and consistent comparison, all experiments were conducted on the Kaggle platform using a P100 GPU runtime. A stratified 5-fold cross-validation approach was used. In each fold, the dataset was split into 80% for training and 20% for testing. This process maintained class distribution. It also enhanced the statistical reliability of the results.

Extensive hyperparameter tuning was performed for each classifier. We used a combination of grid search and empirical testing to find the optimal balance between model accuracy and generalization. The final, optimized hyperparameter values for each classifier are summarized in Table 1. These values were fixed for all subsequent evaluations. This ensured the integrity and comparability of the results across all models.

Table 1: Hyperparameter settings and configurations adopted for all models

| Classifier | Hyperparameter | Optimized Value |
|---|---|---|
| Gradient Boosting | Number of Estimators | 100 |
| | Learning Rate | 0.1 |
| | Maximum Depth | 3 |
| XGBoost | Number of Estimators | 100 |
| | Learning Rate | 0.1 |
| | Maximum Depth | 4 |
| | Eval Metric | $mlogloss$ |
| LightGBM | Number of Estimators | 100 |
| | Learning Rate | 0.1 |
| | Maximum Depth | 5 |

### 3.2 Evaluation metrics

To assess model performance, several complementary metrics were employed. Accuracy measures the overall proportion of correctly classified instances, while precision evaluates the reliability of positive predictions by minimizing false positives. Recall (sensitivity) quantifies the ability to correctly identify true positives, and the F1-score balances precision and recall. In addition, specificity highlights the correct detection of negatives, which is important to reduce false alarms. To ensure robustness, results were reported with 95% confidence intervals (CI), and statistical significance between models was validated using p-values. Together, these metrics provide a comprehensive view of classification reliability.

### 3.3 Results and analysis

To evaluate the performance of the proposed method, three machine learning classifiers were tested using a five-fold cross-validation approach. Evaluation employed four standard metrics: Accuracy, Precision, Recall, and F1-score.

Table 2 summarizes the results of Gradient Boosting, XGBoost, and LightGBM over five folds. For each model, the table lists Accuracy, Precision, Recall, and F1-score per fold, along with the average across all runs. This enables a direct appraisal of each classifier's consistency and predictive strength on the dataset.

Figure 4 shows the confusion matrices for Gradient Boosting, XGBoost, and LightGBM. Comparing the models, Gradient Boosting correctly classified 580 diabetic retinopathy (DR) cases and 568 normal cases, with 20 DR and 32 normal cases misclassified. XGBoost improved on this, correctly classifying 587 DR and 577 normal cases, and misclassifying 13 DR and 23 normal cases. LightGBM performed similarly to XGBoost, with 587 DR and 575 normal cases correctly identified, and 13 DR and 25 normal cases incorrectly predicted. Notably, XGBoost and LightGBM both increased correct DR classification by 7 cases over Gradient Boosting, and XGBoost had the fewest overall

misclassifications. These results allow for a direct comparison of correct and incorrect classifications across the three boosting-based models.

Table 2: Performance comparison of Gradient Boosting, XGBoost, and LightGBM over 5-fold cross-validation.

| Model | Fold | Accuracy | Precision | Recall | F1-score |
|---|---|---|---|---|---|
| Gradient Boosting | 1 | 0.9450 | 0.9334 | 0.9583 | 0.9457 |
| | 2 | 0.9567 | 0.9477 | 0.9667 | 0.9571 |
| | 3 | 0.9533 | 0.9416 | 0.9667 | 0.9539 |
| | 4 | 0.9458 | 0.9294 | 0.9650 | 0.9469 |
| | 5 | 0.9475 | 0.9409 | 0.9550 | 0.9479 |
| | **Avg.** | **0.9497** | **0.9386** | **0.9623** | **0.9503** |
| XGBoost | 1 | 0.9492 | 0.9354 | 0.9650 | 0.9500 |
| | 2 | 0.9700 | 0.9623 | 0.9783 | 0.9702 |
| | 3 | 0.9517 | 0.9385 | 0.9667 | 0.9524 |
| | 4 | 0.9458 | 0.9294 | 0.9650 | 0.9469 |
| | 5 | 0.9467 | 0.9467 | 0.9467 | 0.9467 |
| | **Avg.** | **0.9527** | **0.9425** | **0.9643** | **0.9532** |
| LightGBM | 1 | 0.9517 | 0.9385 | 0.9667 | 0.9524 |
| | 2 | 0.9683 | 0.9592 | 0.9783 | 0.9686 |
| | 3 | 0.9550 | 0.9446 | 0.9667 | 0.9555 |
| | 4 | 0.9508 | 0.9314 | 0.9733 | 0.9519 |
| | 5 | 0.9533 | 0.9564 | 0.9500 | 0.9532 |
| | **Avg.** | **0.9497** | **0.9460** | **0.9670** | **0.9563** |

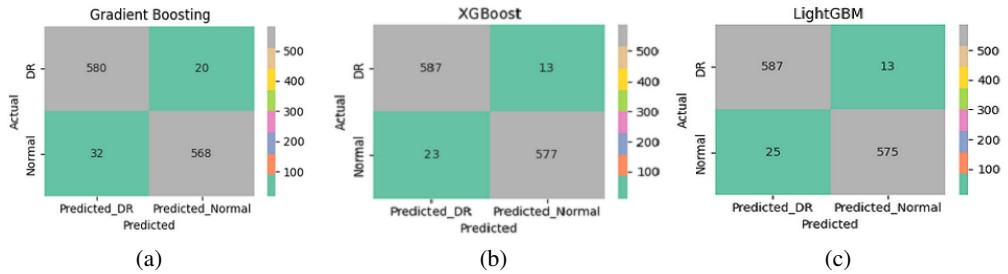

Figure 4: Confusion matrices for the gradient boosting models: (**a**) Gradient Boosting, (**b**) XGBoost, and (**c**) LightGBM.

### 3.4 Discussion

The 5-fold cross-validation results and confusion matrices provide valuable insights into the performance of the boosting-based classifiers. Gradient Boosting, XGBoost, and LightGBM all achieved high accuracy, precision, recall, and F1-scores, confirming the suitability of boosting methods for diabetic retinopathy detection using multifractal features.

Among the evaluated models, XGBoost and LightGBM outperformed the Gradient Boosting model. XGBoost obtained the highest accuracy (0.9527) and recall (0.9643), reflecting strong sensitivity in detecting DR cases. LightGBM achieved the highest F1-score (0.9563) and offered a balanced trade-off between precision and recall, underlining its robustness. Gradient Boosting, while competitive,

exhibited slightly lower metrics and a higher number of misclassifications. These differences highlight the benefits of the advanced optimizations integrated in XGBoost and LightGBM, including regularization, parallelized learning, and leaf-wise tree growth strategies.

Despite these encouraging results, several limitations should be noted. First, the dataset used for training and evaluation was relatively small, which may restrict the generalizability of the models to larger populations. Second, only handcrafted multifractal features were employed, whereas more advanced feature extraction techniques, such as deep learning, were not investigated. Finally, the models were evaluated in controlled experimental settings, and their effectiveness in real-world clinical environments remains to be validated.

Future work will focus on expanding the dataset to improve generalization, as well as exploring hybrid frameworks that integrate deep learning-based feature extraction with boosting classifiers. Combining boosting algorithms with convolutional neural networks could leverage both handcrafted and learned features for enhanced predictive performance. Furthermore, clinical validation in real-world scenarios will be a crucial step toward deploying these models for automated diabetic retinopathy screening.

## 4 Conclusion

This study presented a comparative evaluation of boosting-based classifiers—Gradient Boosting, XGBoost, and LightGBM—for the automated detection of diabetic retinopathy from retinal OCT images using multifractal features. The results demonstrated that all three models achieved strong performance across multiple metrics, confirming the suitability of boosting methods for this task. Among them, XGBoost achieved the highest accuracy and recall, while LightGBM provided the best F1-score and a balanced trade-off between precision and recall. These findings highlight the effectiveness of advanced boosting variants for reliable and efficient retinal disease classification.
Despite these promising outcomes, certain limitations must be acknowledged. The dataset was relatively small, potentially limiting generalizability, and only handcrafted multifractal features were used, excluding deep learning-based representations. Moreover, the experiments were conducted under controlled settings, leaving real-world clinical validation as an open challenge.
Future work will focus on expanding the dataset, exploring deep learning-based feature extraction, and investigating hybrid frameworks that integrate boosting models with neural networks. Validation in clinical environments will also be essential to ensure translational reliability. Ultimately, such advancements could contribute to the development of practical, automated screening systems for early detection of diabetic retinopathy in ophthalmology.

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
