# OpenReview forum: "Boosting-Based Classifiers for Retinal OCT Disease Detection: A Multifractal Feature Approach"
_EurIPS.cc/2025/Workshop/MedEurIPS — EurIPS 2025 Workshop MedEurIPS Submission_

### Official Review · Reviewer_9u7R · 2025-10-25
**Boosting-Based Classifiers for Retinal OCT Disease Detection: A Multifractal Feature Approach**

**Rating:** 5
**Confidence:** 3

**Review:**

This paper introduced a classification method in a retinal OCT disease detection pipeline. The proposed method consisted of multifractal analysis, feature extraction, and gradient boosting classifiers. Although the experimental results were extensive, the introduced solution was not novel. In addition, the lack of comparisons to popular shallow and deep learning methods was a problem.

Pros:
- The description of the method was detailed.
- The paper had a good flow.
- The configurations and experimental results were thoroughly described.

Cons:
- Lack of novelty.
- There were not many baseline methods.
- The paper was too long.

---

### Official Review · Reviewer_pqdx · 2025-10-30
**Limited Novelty**

**Rating:** 2
**Confidence:** 3

**Review:**

The manuscript focuses on the evaluation of boosting-based classifiers—namely Gradient Boosting, XGBoost, and LightGBM—for the automated detection of diabetic retinopathy from retinal OCT images using multifractal features.

However, the paper lacks a novel contribution or any substantive original content. No definitive conclusion regarding the broader research significance can be drawn from this study, aside from identifying which of the three classifiers performs optimally within the limited experimental setup employed by the authors.

Crucially, no appropriate baselines are considered for the Retinal OCT Image Classification task. This makes it impossible to study the benefits of using the multifractal feature representation compared to alternative feature extraction methods.

---

### Decision · Program_Chairs · 2025-10-31

**Decision:**

Reject

**Comment:**

Both reviewers agree that while the paper is clearly written and well structured, it lacks methodological novelty and appropriate baselines.